# The Relationship between Environmental Factors, Satisfaction with Life, and Ecological Education: An Impact Analysis from a Sustainability Pillars Perspective

**Andy Felix Jităreanu** [1], **Mioara Mihăilă** [1,*] , **Ciprian-Ionel Alecu** [2] , **Alexandru-Dragoş Robu** [1], **Gabriela Ignat** [1] **and Carmen Luiza Costuleanu** [1]

[1] Department of Agroeconomy, University of Life Sciences "Ion Ionescu de la Brad", 700490 Iaşi, Romania
[2] "Gh. Zane" Institute for Economic and Social Research, Romanian Academy—Iași Branch, 700481 Iași, Romania
[*] Correspondence: mioara@uaiasi.ro; Tel.: +40-726-061613

**Abstract:** The paper analyzes the link between three concepts: environmental values, ecological behavior, and satisfaction with life. Various studies indicate the links between environmental values and ecological behavior, satisfaction with life, and pro-ecological behavior, but no connection between these three concepts. The paper aimed to develop such a research direction, namely sustainability as an integrative concept. The existence of a complex conceptual model between three specific constructs is analyzed. A questionnaire to 267 students from the universities of Iasi, Romania, was applied. To identify the existence of some equations between components, PLS-SEM and PLS-MGA methods were used, through SmartPLS3.9 and SPSS 18. The final model was a reflective-formative one on two hierarchical levels, being made up of 5 latent variables and 18 observed ones. Only the existence of significant equations between knowledge of environmental values and pro-ecological behavior and between knowledge of environmental values and satisfaction with life was highlighted. The multi-group analysis: although higher values are obtained among students from the rural compared to urban, there are no significant differences. The influence of knowledge of environmental values and ecological behavior on life satisfaction among students was highlighted. The results encourage the continuation of research on a larger population, from an extended area.

**Keywords:** natural resources; conservation; behaviour; involvement; life satisfaction; pillars; sustainable development

## 1. Introduction

The issue tackled in this paper starts from the ever-stronger demands of ensuring a sustainable increase of satisfaction with life. The basic concept and context taken into account for the present research is that of sustainable development. The particular concepts that determine the approach and the construction of the paper models are: environmental values, ecological behavior, and life satisfaction. The approach, linking the three concepts in a comprehensive whole, is based on the close interdependence between the knowledge regarding the sustainable development and the specific achievement actions. All these concerns are of major interest and up to date for the international policies as well as for the individual activities of people. The welfare and life satisfaction are aspects that the individuals take more and more into account in order to raise the satisfaction with life.

The context of the dependence of life satisfaction on the requests and objectives of sustainable development is given by: the growth of pollution level, the increased environmental degradation, the concern about the depletion of natural resources, the inadequate eating habits that generate health problems, the protection of the natural resources, and the preoccupation for achieving a high level of emotional welfare.

The present paper focuses on the link analysis between the environmental values, the ecological behavior, and the individuals' life satisfaction. The application, based on the survey method, is done on a sample of 267 students from Iași academic centre: "Ion Ionescu de la Brad" University of Life Sciences, "Alexandru Ioan Cuza" University, and "Gheorghe Asachi" Technical University. The decision of choosing students as the research universe is supported by recent studies that indicate the fact that concerns regarding the natural resources and man's relationship with the environment are significant among young people [1,2]. Moreover, the aspects related to the knowledge of environmental values and to assuming an ecological behavior are significant for the age groups that manifest a high level of responsibility by virtue of maturing and who are enrolled in education and learning systems [3–5]. The performed research focuses on the analysis of the influence that the taxonomic knowledge of the environmental resources and the value reference to them has on the emotional state of the individuals and on their life satisfaction. In fact, it starts from the general hypothesis supported by the specialised literature that the interest for the knowledge of environmental values and the adoption of "pro-eco" behavior are determining factors with regard to life satisfaction [6,7]. The variables of the scales used in the investigation are: the natural resources, the environmental values associated with notions of ecological values that include ecological behaviors as well, the knowledge of environmental values, and life satisfaction. The linking element among the notions and concepts is the framework of sustainable development goals, which involves assuming an ecological behavior. However, in order to attain this goal, knowing the environmental resources and appreciating them are necessary.

A fundamental supporting element of this paper is the context of orienting the international and national policies of supporting sustainable development, which involves the acceleration of assuming a general ecological behavior. Regarding the SDG-17 sustainable development objectives referring to sustainable production and consumption, together with the environmental protection, the research takes into account work variables such as: the natural resources, the knowledge of their value, the specificity of the behavior towards the environment and the individuals' satisfaction with life. As conceptual theoretical support, established scales were used for measuring these variables: NAVS (Natural Area Value Scale) and life satisfaction (SWLS). To these, we have added a scale constructed and adapted according to validated scales in order to analyze the ecological behaviour. The scales are described in the methodology chapter.

The main gaps from the specialized literature: the relationship between concepts has been partially or particularized emphasized, such as: between environmental values and pro-ecological behavior or between life satisfaction and pro-ecological behavior, with an emphasis on gender or training differences; the issue of environmental values is predominantly approached from an ecological perspective, a fact that maintains confusion regarding the concept of sustainable development or assessment of the value of environmental resources; the issue of sustainable development still remains a predominant theoretical one, the consequences of sustainable development requirements on individual lives being too little emphasized; knowledge of environmental values is not strongly anchored in the individual life and there is no framework to create the universe of knowledge of environmental values.

Thus, the research has an interdisciplinary perspective, even if apparently subjective, as the concept of satisfaction with life. We highlighted the significance of ecological behavior and knowledge of environmental values for life satisfaction among students. Through the emphasis of interdisciplinarity and methodological borrowing, the interdisciplinary analysis of a new conceptual model was followed to capture the existence of the link between the knowledge of environmental values, pro-ecological behavior, and life satisfaction among the student population in Iasi.

For this reason, the basic concepts analyzed in the paper are presented according to literature and the state of the art for his topic:

1.  Ecological behaviour concept is much more present in the literature, at least compared to the concept of environmental values.



From the point of view of practically taking into consideration the concepts and the results of the paper, these are highlighted in the ecological behaviour component, which can be measured practically by specific actions such as: the consumption of ecological products (food and non-food), the use of ecological means of living (transport, recycling, saving resources), and availability in investing time and/or money in order to learn and practice an ecological behaviour. The components of environmental values and life satisfaction, although predominantly subjective in nature and difficult to measure directly, are factors that determine ecological behaviour. In the Eurostat reports, where the environment is approached within the context of sustainability, it is mentioned: "the environmental conditions not only affect human health and well-being directly, but also indirectly, as they may have adverse effects on ecosystems, biodiversity, etc." [8,9]. It is important to mention that the environment is an independent factor that affects the quality of life, but the health behaviours, including the behavior toward the environment, affect human satisfaction with life [10,11].

The present work aims to contribute to the support and completion of the literature through which the interest in the analysis of the connection between the environmental issue and that of the satisfaction with life will be a common one. Thus, we want to demonstrate that the problems of the environment, seen as an external frame of reference, and those of the individual (health, quality of life, life satisfaction, behaviors, etc.) can no longer be approached as independent. The main aim of the work is to highlight the influence of environmental values on the satisfaction with life and vice versa.

The pro-environmental orientation is also a pillar of the modern culture through the consolidation of the socio-structural factors that make people form environmental values, attitudes, and behaviours [12,13]. In this regard, the European policies, too, are directed towards the support of the ecological behaviour. The European Green Deal or Organic Action Plan are an integral part of the general policy of supporting sustainable development, which is one of the most important objectives of the European Union (EU) [14–16]. The European Green Deal is one of the greatest action plans for the support of sustainable development by assuming an ecological behaviour of "A new Circular Economy Action Plan" (CEAP) of the European Commission.

In this context, consumption behaviour is considered as a tool to demonstrate the people's care and interest for the natural resources and environment. The sustainable consumption, as a fundamental objective of the European policies of sustainable development, cannot be supported without the "pro-eco" behaviour or without the minimum value recognition of the natural resources. Although the roles of consumption and lifestyles have been acknowledged in some EU-wide policy initiatives and certain progress has been made, many challenges remain to achieve more sustainable consumption. A number of factors have been identified as influencing people's predisposition to sustainable behaviour, including convenience, force of habit, availability, affordability, and product performance [1,17]. The Action Plan for Sustainable Consumption and Production and Sustainable Industrial Policy of the European Commission (2008) debates the sustainable consumption and consumer behaviour as well, as a significant challenge for society nowadays. In this regard, in 2011 the study "Policies to promote sustainable consumption" was elaborated to determine the interest for this orientation [18]. Overall, the sustainable consumption should not be confused with the ecological concerns, because sustainability is about economic growth and social progress at the same time. So, the physical and emotional health is in line with the sustainable consumption [18–20].

It is true that the notion of ecological behaviour is vast and includes numerous references from the perspective of measuring this type of behaviour. Thus, "the ecological behaviour does not refer only to the consumption of certified ecological foods, a phenomenon that has generated many changes in food industry, but also to actions oriented towards the protection of natural resources or the generation of a small quantity of waste." [21]. The theories according to which behaviour is influenced by values are a basis of this research [22], and Kasser debates about the empirical studies of the relationship between consumption

and wellbeing, accentuating the idea of sustainable consumption negatively linked to the increase of the quality of life. He found the positive association in correlational studies considering the hedonic, cognitive, and eudaemonic dimensions of wellbeing [23].

For these reasons, the policies for sustainability consider the engagement of citizens to recognize and appreciate the intrinsic and extrinsic values of the natural resources before manifesting an ecological behavior. And in this respect, the debates from the literature and from practice point out the differences between the consumers' "wants" and "needs" [18]. This topic derives from the finding that the intentions and thoughts are not always materialized in actions. In accordance with the EU reports, "faced with a global scarcity of natural resources, 'doing more with less' has become the main challenge for producers and consumers" [16]. This announced a sustainable product policy legislative initiative to make products fit for a climate-neutral, resource-efficient and circular economy; reduce waste; and ensure that the performance of frontrunners in sustainability progressively becomes the norm.

2. Environmental values concept

A fundamental pillar of the sustainable development is the ecological behaviour which, in turn, is based on the knowledge and the appreciation of environmental values. Environmental values refer to the complexity of the natural resources that people appreciate, evaluate, and use in order to increase the satisfaction with life and of the human development [24–26]. "All products have a natural basis. The EU's economy is highly dependent on natural resources" [27].

The preoccupations for the transition to sustainable development by means of an ecological behaviour as well are present in the international policies and in the specialized literature. Between the two spheres of interest there is a direct connection given by the responsibility and accountability of social groups of supporting this orientation in the conditions in which "a sustainable consumption behaviour represents a fundamental element of the long term development" [28].

According to consecrated literature, four types of values are very relevant in predicting environmental beliefs and behaviours [29–32]: biospheric (as valuing the environment), altruistic (as valuing the welfare and wellbeing of other human beings), egoistic (as valuing personal resources), and hedonic values (as valuing pleasure and comfort). These values are measured with a validated scale, the Environmental-SVS (E-SVS), adapted from Schwartz Value Survey (SVS) [29,33].

The approaches of the topic related to environmental values are various depending on the research aim or the socio-demographic dashboard and the citizens' purposes in life. Some surveys ask participants how ecosystems matter to their livelihood and wellbeing; others involve experts rather than citizens to scientifically quantify the value of ecological processes [34]. Therefore, the debates about environmental values and valuation are perplexing, in part because these terms are used in vastly different ways in a variety of contexts [34].

Specialised literature and studies show that the natural resources are the linking element that connects the global environmental and natural resources aspects with individual aspects related to life satisfaction. The specialised literature abounds and is demonstrative in this respect.

In order to confirm the link between the parameters for evaluating environmental resources and those for evaluating the psycho-emotional status of young people, recent international studies (conducted among young people between the ages of 16 and 25) regarding eco-anxiety were identified. They show that eco-anxiety, or concern for the state of the environment, "profoundly affects a large number of these young people around the world, and that ignoring this new condition can lead to health risks and social inequalities." [35–37]. Grose indicates the health impact of experiencing eco-anxiety in a practical manner as: why it is important to reduce plastic waste, which is a more effective way to bring the carbon footprint down or to adopt a pro-environmental behavior [38].

According to NAVS (Natural Area Value Scale) of Winter & Lockwood, used for this research as a principal support, and which considers the concept of knowing the environmental resources, the value taxonomy of these ones is built based on four categories of latent values [39]:

- intrinsic individual values
- non-use values (of conservation)
- use values (exploitation)
- recreational values: regarding this category, the direct connection to the concept of welfare can be identified.

The issue of the value appreciation of environmental resources is rather controversial, some authors being the supporters of the generally valid attitude towards the environment, whereas others assert that this cannot be one-dimensional: "the beliefs in such a complex domain as ecology are not simple, but complex and multidimensional" [40]. This thesis is functional nowadays and represents a foundation of the environmental policies to which the individuals should relate with more responsibility and conviction. In this regard, there is literature similar to the interest of this research. As part of a study based on a survey, Chan interviewed 992 students in Hong Kong using the Weigel & Weigel scale regarding the concerns related to the environment with the aim of investigating the availability to engage in various pro-environmental behaviours, including common norm behaviours: paper recycling, using less supplies, plastic bags, etc. [41]. On the other hand, the satisfaction with life is the cognitive dimension of the hedonic approach to wellbeing. In fact, it is a global assessment that the individual makes with regard to his own life [42,43].

The circular relationship between knowledge, behaviour, feelings, and values can be coordinated by means of adequate education that is accomplished differently on four levels [44,45]:

- the level of knowledge: understanding the way the environment functions, the causes of problem occurrence, the ways of solving them, and its role in man's life;
- the level of skills and competences: efforts directed towards the training, development, and valorization of the skills necessary for the identification of the situations nature faces;
- the level of emotions: actions directed towards the acquisition of a set of values concerning the environment as well as the stimulation of the motivation of participating in activities meant to improve its state;
- the level of behaviour: the training and the orientation of the personal capacity towards using knowledge with the purpose of making positive and correct decisions related to the environment.

3. Satisfaction with life concept

The current reality of the context in which the individual leads his life, mainly measured by means of the socio-economic welfare, significantly affects the wellbeing, the personal actions, and the individual's beliefs and attitudes [46,47]. "From the ecological point of view, at various levels, the society in which we live is inefficient, immoral, unhealthy, unfair, and affects the sustainability of the natural habitat", and from this perspective it is more than necessary to proceed to training, learning, and knowing the environmental values [21]. In this context, the notion of ecological citizenship is taken into consideration. This notion was put forward by Christoff and pursues the directions of moral excellence ("a fair distribution of the ecological space"), duty, and responsibility ("the obligation of ensuring the sustainability of activities") [48,49]. Seyfang tested the hypothesis according to which the ecological citizenship represents a driving force for the sustainable consumption, by means of the ecological behaviour of the consumers (for example, encouraging the purchase of more food products produced locally) [50].

"More recent thinking suggests that the decision to adopt appropriate pro-environmental behaviors will reflect some general psychological orientations, or values, that are culturally patterned" [51]. Moreover, there are papers that planned and built value sets and models

to explain the relationships among values and wellbeing, and to demarcate values, desires, and personality [52]. "Environmental wellbeing is about leading a lifestyle that values the relationship between ourselves, our community, and the planet. It's on all of us to live in harmony with the earth, recognize our impact on the environment, and promote practices that sustain Earth's resources." [53]. What is more, "considering well-being at the ecological scale may call to mind degraded places and endemic ecological challenges that impact human and environmental health. Environmental crises provide opportunities to address the root causes of on-going and acute environmental injustices" [54].

A considerable part of the specialised literature is dedicated to the effect of the individuals' pro-environmental behaviour on their satisfaction with life [55,56]. Nevertheless, the attention given to a reversed relation, i.e., the effect of life satisfaction on the pro-environmental behaviours, has been limited. The results of the studies performed in this respect indicate that people's life satisfaction stimulates their interest of participating in a pro-environmental behaviour. Consequently, an individual's preoccupation with the environment represents a primordial mechanism for someone's life satisfaction and in turn the level of people's life satisfaction exerts a considerable influence on the pro-environmental behaviour. Thus, improving the satisfaction with life for the general public may turn into a spontaneous instrument of solving potential conflicts between the economic growth and the environmental protection [57]. On population samples from Canada and the United States there have been done studies that examined the way in which a variety of pro-environmental behaviours (PEB) have predicted satisfaction with life. The hypotheses of these studies were: the perception of ecological threats, the care for the protection of environmental resources, and a more frequent involvement in pro-environmental behaviours, which predicted a greater life satisfaction. The results indicated that PEB were positively correlated with individuals' life satisfaction. In addition, life satisfaction correlated more strongly with behaviours that involved more social interaction and behaviours that involved direct costs in terms of money, time, and effort. Besides, the perceptions regarding the concerns related to environmental problems negatively predicted life satisfaction. Therefore, people who are more concerned regarding the problems they notice with respect to the natural resources are less satisfied with life. Nevertheless, it has been observed that this category of individuals is more likely to engage in pro-environmental behaviour, which supports the achievement of sustainable development goals [58]. The conclusions of the studies that have approached the connection between the environmental values and the satisfaction with life are that the environment is a cross-cutting factor crucial to wellbeing [59].

The purpose of the paper is to demonstrate the existence of the link between the environmental values, the ecological behaviour, and the individual's satisfaction with life. The main pursued objectives are:

O1. The analysis of the influence of knowing the environmental values on the wellbeing of the individual;

O2. The identification of the influence of social differences on the ecological behaviour;

O3. The analysis of the ecological behaviour from the perspective of its impact on the wellbeing.

The secondary pursued objectives are:

O4. Creating and using the appropriate interdisciplinary research framework, by considering the two fundamental issues: the environment with implicit natural resources and life satisfaction from the perspective of physical and emotional health.

O5. The positioning on a higher level, on a scale of institutional and individual interest, of the subjects of this research.

O6. The proposal, based on the results, of several recommendations and measures that can be adopted to support a more dynamic "pro-eco" behavior.

The proposed objectives are formulated as a result of some assumptions, which started by analyzing the literature. The idea of knowing the environmental values among students

is developed, as well as the influence of this and ecological behaviour on the satisfaction with life. So, the paper highlights the connection between these topics—environmental values, pro-environmental behavior, and life satisfaction—and reduces the analysis to a common basic element: the importance of knowing environmental values. To attain the research objectives and perform the market investigation, we used established scales and adapted some scales in a construct specific to the objectives of this research topic. Achieving the proposed objectives is possible by structuring the work according to a carefully organized plan. Thus, in the first stage, a selection of specialized literature is highlighted and it confirms the importance of the theme and the interest in it. Then, the work methodology is built and presented, which starts from a set of useful and necessary variables in the questionnaire carried out on a sample of young students defined in accordance with the purpose of the work. The hypotheses formulated are the basis of the actual analysis and are supported by the relevant literature. The statistical analysis, carried out with specific programs, leads to a set of results that are used both for the validation of the achievement of the objectives and for the recommendations for giving more interest to such a research subject.

These are significant considerations for which, by means of this research, a connection was conducted between the European environmental policies, the pro-eco behaviour, and life satisfaction. In fact, without an adequate policy, the pro-eco behaviour cannot be stimulated, and pro-eco behaviour has a great chance of contributing to the increase or the improvement of the individual's wellbeing. To correlate the global interests of protection and valorization of environmental resources with the individual's life quality, ecological behaviour proves to be a successful binder. Some studies manage the relationship between sustainable consumption practices and wellbeing with the possibility to decide the measures for the human wellbeing and environmental sustainability, too [60]. Approaching the relationship between the issue of environmental resources and that of life satisfaction, both complex and generous, involves a consideration of value. So, in order to identify the nature of the connection between these themes, it is indicated to aim the analysis towards a value approach, of the "how much does it matter" type.

The results of the work can enrich the literature, especially domestic, and can determine the interest of the study for a new issue: the importance and role of knowledge of environmental values among young people, from the perspective of life satisfaction; decisions can also be triggered to introduce into the educational system some programs that include knowledge of environmental values. Because for now, a view of the interconnection of two different domains—the environment and the emotional state of individuals—is limited, through the formulated objectives, to the analysis of the specialized literature and the paper development; the essential contribution was to unite the concepts belonging to these different fields. In addition, the novelty of the paper consists in the validation of a complex formative-reflexive conceptual model based on three distinct constructs. In this model, an interdisciplinary approach is emphasized through the connection of two apparently independent fields: environment and psychology.

## 2. Materials and Methods

### 2.1. Methodology

The present research is of quantitative nature and in order to attain the proposed objectives, methods of gathering data though a survey were resorted to, whereas for the exploratory processing, modelling through structural equations using iterative algorithms based on the partial least squares method was used (PLS-SEM) [61]. The gathering of data for the quantitative research was done by means of a questionnaire as a research instrument (available in the Annex) with 35 specific items, applied in Iași university centre between March and April 2022, among the students from three universities: "Ion Ionescu de la Brad" University of Life Sciences, "Alexandru Ioan Cuza" University, and "Gheorghe Asachi" Technical University. Non-probabilistic sampling was used through a mixture of snowball technique and self-selection techniques.

The research has been conceived so as to evaluate, among the students in Iași, the way in which these ones interact directly and indirectly with theoretical and practical elements regarding the ecological environment and the way in which personal wellbeing was being reflected. Taking into consideration exploratory research contexts in which a large amount of data is combined with structural theoretical concepts, data processing using iterative algorithms based on the partial least squares method for modelling structural equations using Smart-PLS software was utilised [62].

The fundamental pillars of the research are the following three:

- The European sustainability and natural resources protection policies and strategies;
- The ecological behaviour in the context of the demands for attaining the objectives related to these strategies;
- The life satisfaction of individuals as an indicator of the quality of life, which is an objective of sustainable development.

A main direction pursued through research was the identification of a common reference point of the three pillars.

The following are the working tools that formed the basis of the organization of the research.

To develop the research tool—the questionnaire—a set of appropriate, established, and validated scales on the researched topic was considered. Then, we adapted another set of scales to the research objectives: Natural Area Value Scale (NAVS), New Ecological Paradigm (NEP), Two Major Environmental Values (2-MEV), Thompson and Barton's ecocentrism–anthropocentrism scales, Environmental Portrait Value Questionnaire (E-PVQ), the General Ecological Behavior (GEB), The Student Life Satisfaction Scale (SLSS), and Multidimensional Student Life Satisfaction Scale (MSLSS).

The reference scale used in developing our own tool for researching the behaviours towards the environmental resources, from the perspective of the connection with the welfare, is NAVS scale. This scale with 20 assertions appreciates the environmental values from the perspective of knowing and recognising them and is used in the first part of the questionnaire. The structure of the established NAVS scale takes into account the taxonomy of the environmental values: intrinsic values, use values, recreational values, and non-use values (non-material). To build our own questionnaire—in accordance with the proposed topic—we also used Satisfaction with Life Scale with five items [63] according to Fetzer Institute.

NAVS Scale (Natural Area Value Scale) is one of the best known and used scales for the psychological appreciation of the environmental values [39]. In the paper "A model for measuring natural area values and park preferences" the scale is described and presented in detail, making reference to the theories that suggest that the value appreciation of an asset, of a resource, is essential in determining the individual behaviours regarding the issue of the environment. We used the results of the mentioned paper as a reference model that explains the empiric relationship between value and behaviour. This fact supports the opinion from the present paper that the more appreciated an asset or a resource is—by means of a value judgment—the more the behaviour towards it is a "pro" one. The NAVS scale describes in fact, in a clarifying manner, the relationship between values and behaviours from the perspective of perceiving natural resources. The expressed hypothesis in building the NAVS scale has been that the relationship between the "pro" environment values and behaviours are based on theoretical and empirical papers, with the theories proposed by Stern & Dietz and by Lockwood being very suggestive [29,64].

Weigel and collaborators [65] developed an instrument of measuring the general attitude of the population regarding the environment: scale of environmental concerns, with 17 items that refer to pollution, natural resources conservation, biodiversity protection, the optimism regarding the environment, and the impact on one's own person. Arbuthnot developed the scale of attitude towards the environment, with 24 items that measure the attitude towards ecological aspects from the economic, legislative, technological, environmental power, and future orientation points of view [66]. Dunlap and Van Liere developed

the scale of the new paradigm on the environment-NEP (New Environmental Paradigm), with 12 items that measure the respondents' attitudes regarding man's influence on nature, the limits of population growth, and whether people should have the right to dominate and rule over nature [67]. In the studies developed by Arcury, it is demonstrated that the NEP scale correlates positively with the level of knowledge the individuals have regarding the environment [68,69]. NEP was tested in Hong Kong among students [70]. To these are added other studies that pursue the predictability regarding the availability to engage in various pro-environmental behaviours. A conclusion of the study in Hong Kong indicates extremely positive attitudes towards the environment, especially with regard to the protection of wild animals and the conservation of natural resources. It is essential to mention that the authors of the research insisted that the government regulations regarding pollution be updated and put into practice and that people should participate in pro-environmental activities. A significant positive relationship between the students' attitudes and their willingness to engage in various pro-environmental behaviours was identified.

Satisfaction with Life Scale (SWLS) with large applicability is a valid and reliable instrument for assessing satisfaction with life in diverse population groups [71]. In a 5-item list, the participants indicate how much they agree or disagree with each aspect using a 7-point scale that ranges from 7—strongly agree, to 1—strongly disagree. This scale has favorable psychometric properties, high internal consistency, and high temporal reliability, being adequate for use in different age groups and with potential for other interdisciplinary uses [63].

By continuing the process of correlating the analyzed concepts, it arrived at the notion of behaviour, which sums up, in fact, the willingness of the individual to making personal sacrifices or investments in order to protect the resources that belong to the environment. We mention that in the research, emphasis is laid mainly on the intrinsic behaviour, which pertains to one's own will. The resulted behaviour of some constraints or other forms of external determination (such as punishments, fines, rewards, social approval or disapproval etc.) is not taken into consideration in the present paper.

The issue of environmental values is a relatively subjective one, and the literature dedicates interest to define these as clear as possible. In this regard, some studies suggest this classification related to environmental resources: [32]

- Biospheric Values (Bio):
  - ○ Bio1: protecting natural resources, being important to prevent environmental pollution;
  - ○ Bio2: preserving nature, being important to protect the environment;
  - ○ Bio3: harmonising with other species, because it is important to respect the nature;
  - ○ Bio4: fitting into nature and being in unity with nature;
- Hedonic Values (Hed):
  - ○ Hed1: pleasure and gratification of desires, to have fun;
  - ○ Hed2: enjoying food, leisure, and life's pleasures;
  - ○ Hed3: self-indulgence, doing pleasant things;
- Egoistic Values (Ego):
  - ○ Ego1: social power, dominance, control;
  - ○ Ego2: authority, right to command, to lead;
  - ○ Ego3: influential, impact on people or events/actions;
  - ○ Ego4: material possession, money, wealth;
  - ○ Ego5: hardworking, aspiring, ambition.

For the survey, we elaborated the structure of the questionnaire on three components supported by associated concepts. Reference has been made with regard to the following aspects, identified in the specialised literature and completed in accordance with the research objectives:

1. Knowledge:
   - knowledge of environmental values: learning, acquiring, gathering, and being accountable regarding the information related to environmental values from a rather scientific perspective;
   - the taxonomy of environmental values: categorizing the values and the information related to the environment based on diverse personal, cognitive, social and criteria, etc.;
   - issuing value judgments for natural resources and appreciating their role and the importance for the daily life;
2. Behaviour:
   - the benefits received from the relationship with the environment: identifying, expressing, and capitalizing on the benefits and the opportunities that may be derived and used in relation to the environment;
   - the manifested behaviour towards the environment: willingness to invest time and/or money in recognising and appreciating the environmental values;
3. The level of satisfaction with life: identified through established scales and derived from the pro-eco behaviour.

### 2.2. Research Design

The formulated statistical hypotheses are the following:

**H1.** *Knowledge of the environmental values (NAVS) has a significant influence on the pro-ecological behaviour (PROECO) among the students.*

The knowledge of the environmental values is based on the taxonomic knowledge of natural resources, which is acquired through education, models, and personal documentation effort. However, pro-eco behaviour or behaviour supporting the environmental values without a minimum level of knowledge is almost impossible to manifest. The individuals that support pro-eco behaviours only through self-determination, without knowledge, are limited in number. Students are supposed to benefit greater chances of knowing the environmental values and of adopting pro-eco behaviour, first through the educational programs which, in turn, are based on sustainable policies and strategies. Therefore, we assume that the students who acquired—through conviction or learning—more knowledge of the environmental problems and values have greater chances of acquiring and manifesting pro-eco behaviour, even in the long run.

**H2.** *Knowledge of the environmental values (NAVS) has a significant influence on life satisfaction (LSF) among the students.*

This is a hypothesis that may prove a bit bold, but is based on the results from specialised studies. In the context in which the social dynamics are more emphasized, the level of stress is greater, and the performance requirements are more advanced, students face problems of understanding and measuring life satisfaction. In the conditions in which more individuals return to the environmental resources as a means of reducing stress and ensuring life satisfaction, we assume that this orientation towards the environmental values is valid for students as well.

**H3.** *Between assuming a pro-ecological behaviour (PROECO) and life satisfaction (LSF) there is a significant relationship among the students.*

In specialised literature, the connection between pro-eco behaviour and life satisfaction is demonstrated in both directions: pro-eco behaviour stimulates life satisfaction and life satisfaction determines and stimulates pro-eco behaviour. Through this hypothesis, we attempt to analyse these directions of influence for the Romanian students as well.

**H4.** *Assuming a pro-ecological (PROECO) behaviour among the students has a significant impact in the relationship between the knowledge of environmental values (NAVS) and life satisfaction (LSF).*

Based on the previously assumed hypotheses, it is naturally logical to assume the fact that people may reach a superior welfare when the two dimensions: knowledge of the environmental values and the pro-ecological behaviour, are a natural and constant part of students' life. Consequently, it is normal to appreciate that starting from the knowledge of values by means of a specific behaviour, we may obtain a relationship with the variable of life satisfaction that is superior to the situation in which connections would be treated separately.

**H5.** *There are no significant differences within the researched structural model between the students from the rural area and that of the students from the urban area.*

Since the specialised literature attaches significant importance to the urban-rural differences with regard to the attitudes, behaviours, and perceptions related to the environmental values, we support this hypothesis. Literature has demonstrated that the problem of these differences is sustained, of interest, and up to date. The importance of education in attaining some sustainable goals in the long run demands the analysis of the differences among groups based on age, level of studies, sex, and background, which are frequently analyzed in the present research [72–74].

Another perspective from which the present research has been performed is that of the socio-demographic differences as influence on the pro-eco behaviour. We considered the urban-rural differences and the male-female gender differences in agreement with the specialized literature. We started from the conclusion that the differences between the rural and the urban population are well supported and documented in the literature regarding the issues of the environment and the natural resources. Thus, the background, rural or urban, may exert different influences on the interest of assuming pro-eco behaviours, as well as on the diverse forms of concern regarding the state of the environment. A set of studies carried out in Canada show that, based on the cognitive values (the taxonomy of the environmental resources, their value appreciation, the attitude towards the environment) and of some behaviour indicators, the differences of rural-urban may be explored. The results showed insignificant differences between rural and urban regarding the indicators of the ecological behaviour. However, the inhabitants from the rural area obtained a higher score with regard to the altruistic values, gave greater priority to the environment, and reported a more increased participation in recycling actions and pro-environmental specific behaviour [75].

Specialised studies have analysed comparatively the values, attitudes, and environmental behaviours of some samples from the rural and urban areas. The New Ecological Paradigm (NEP) scale was used, a specially conceived scale of moral duty and a scale of pro-environmental behavioural intentions. The results indicate high levels of concern for the environment and low levels of pro-environmental behaviour in both samples. Comparing the two samples, it was observed that those who live in the cities assume a greater number of environmental responsibility values, but manifest less pro-environmental inclination when the attitude and behavioural intention scales are used. People living in the rural area present more focused attitudes of responsibility towards the environment and a greater consistency in expressing behavioural intentions compatible with environmental protection [12]. Other studies have indicated that pupils living in the urban and suburban areas manifested stronger verbal engagements towards the pro-environmental behaviour compared to pupils from rural areas, taking into account the reasons of these differences as well as the consequences for the educational approaches [76] (Figure 1).

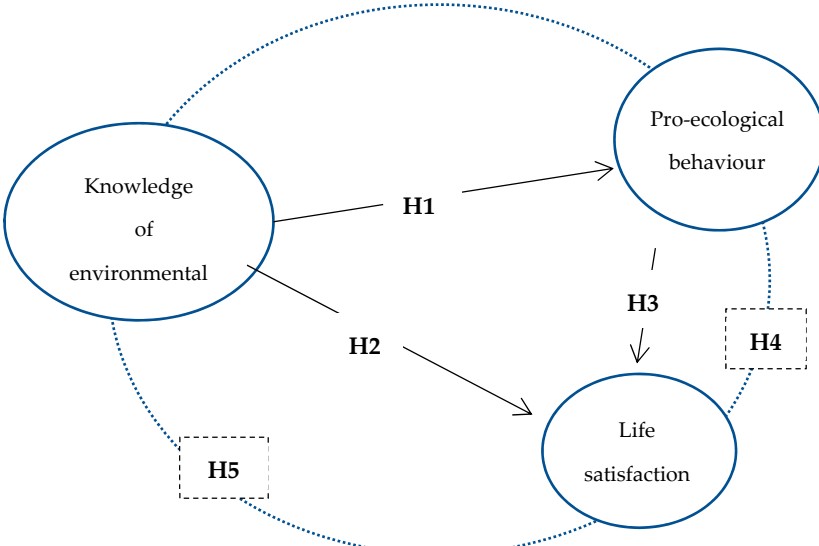

**Figure 1.** Research design.

To achieve the proposed objectives and test the research hypotheses, a survey was carried out between March and April 2022, among the students from three universities in Iași, which have specializations related to natural resources and environmental values: "Ion Ionescu de la Brad" University of Life Sciences (The Faculty of Agriculture), "Alexandru Ioan Cuza" University (The Faculty of Geography), and "Gheorghe Asachi" Technical University (Faculty of Chemical Engineering and Environmental Protection).

The questionnaire was conceived to evaluate, among the students, the way in which these interact directly and indirectly with theoretical and practical elements regarding the development of a sustainable environment.

The present research used a non-probable sampling method with voluntary response, starting from the snowball principle and self-selection techniques and mainly relying on students' volunteering and the facility of access in the process of completing. Each respondent agreed in advance to participate in the study and to complete the questionnaire. The design of the questionnaire was structured in four sections. The first three referred to the constructs: knowledge of environmental values (20 items), pro-ecological behaviour (10 items), and life satisfaction (5 items). The last section took into account the socio-demographic dimensions of the sample (gender, age, specialization, and year of study).

The questionnaire pretesting was done on a small sample batch: 36 respondents. The aspects that were observed referred to the filling-in convenience, checking the adequacy of the questions and managing potential errors. The respondents were asked to answer using a Likert scale of five points: where 1 = total disagreement, 2 = disagreement, 3 = neither agreement nor disagreement, 4 = agreement, 5 = total agreement. The appropriateness of questions was highlighted and no subsequent pretesting change was necessary.

The final batch comprised a number of 267 valid questionnaires filled in by the students from the 303 questionnaires applied (88.1%). The volume of the batch was sufficient for achieving the objectives of the assumed exploratory research (the necessary minimum being of 35 questions × 5 = 175 respondents).

Knowledge of the environmental values is the first aspect that directly converges towards assuming the understanding of the concept of sustainable development and was analyzed by means of the NAVS questionnaire made up of 20 items. Essentially, these have different attributions in accordance with the personality structure, personal interest, or the individuals' life purpose. For this reason, the four established categories of knowing the environmental values were used: intrinsic values (NAVS1-NAVS6), use values (NAVS7-NAVS12), recreational values (NAVS13-NAVS14), and non-use values (non-material) (NAVS15-NAVS20). The pro-ecological behavior was described in the second

section of the questionnaire and was structured in three subcategories: attitudes—4 items (from PRO_ECO1_1 to PRO_ECO1_4), actual behaviour—4 items (from PRO_ECO2_1 to PRO_ECO2_4) and action: 2 items (PRO_ECO3_1 and PRO_ECO3_2). The questionnaire regarding life satisfaction comprised 5 items (SLS1–SLS5) and reflected the students' well-being or how satisfied they are with their life. The socio-demographic results highlighted that from the 267 student respondents from Iași University of Life Sciences, 78.3% were women and 21.7% men (Table 1).

**Table 1.** Socio-demographic characteristics of respondents.

| Criteria | N | % |
|---|---|---|
| Sex | | |
| Male | 58 | 21.7 |
| Female | 209 | 78.3 |
| Background | | |
| Urban | 149 | 55.8 |
| Rural | 118 | 44.2 |
| Year of study | | |
| Year 1 | 108 | 40.4 |
| Year 2 | 79 | 29.6 |
| Year 3 | 61 | 22.8 |
| Master | 18 | 6.7 |
| Specialisation | | |
| Specialisation 1—Consumer and Environmental Protection | 35 | 13.1 |
| Specialisation 2—Agriculture and Montanology | 74 | 27.7 |
| Specialisation 3—Environmental Geography | 79 | 29.6 |
| Specialisation 4—Environmental Engineering | 78 | 29.2 |
| Age | | |
| Min = 19 | Max = 47 | Average = 21.22 |

Source: Data processed by the authors using SPSS Inc. Released 2009. PASW Statistics for Windows, Version 18.0. SPSS Inc.: Chicago, IL, USA.

About the structure of the university learning system in Romania, its functionality is based on the Bologna System, implemented by all the Romanian universities, according to the principles of the European higher education area. So, the functional university structure is currently divided into three stages of the cycle higher education: bachelor's (3 years and just for some exceptions 4 years), master's (2 years), and doctoral studies (3 years).

The student respondents have ages between 19 and 47, with an average of 21.22 years. Most are 21 years of age, 99 students respectively representing 37.1% of the respondents. The greatest majority are in their 1st year of study (40.4%) and only 6.7% are enrolled in master studies. Regarding their background, 55.8% of the respondents come from the urban area and 44.2% from the rural area.

During the application of the questionnaire, there were discussions with the students from Life Sciences University, regarding various aspects of pro-ecological behavior and the knowledge of environmental values specific to their field of study. The most relevant are: the consumption of alternative fuels [77] and consumption of non-genetically modified products. Similarly, the students from Iași Technical University brought into discussion the need for other questions to be added to the questionnaire, which would include elements regarding the sustainable dimension of consumption or such as the valorisation of waste [78], as well as obtaining energy from alternative renewable sources [79,80]. The students from economics emphasized the valorization in the questionnaire of the native anthropic and

cultural dimensions, as well as the resources as specific elements of sustainability. It is thus necessary to diversify the content of the questionnaire regarding the knowledge of environmental values at a much more complex level, as well as the questionnaire regarding the ecological behavior of the consumer regarding the new renewable technologies.

### 2.3. Methods

The research used a questionnaire with 35 items applied among students from Iasi, from 3 different universities. Two-hundred-and-sixty-seven valid answers were obtained and processed with the help of SPSS 18 and SmartPLS 3.9.

The partial least squares PLS-SEM and PLS multi-group analysis (MGA) methods were used to validate the conceptual model and research the advanced hypotheses. Consequently, the initial model analyzed with the help of SmartPLS 3.9 software had the following structural form described in Figure 2, with 35 reflective variables, 8 reflective latent variables, and 2 formative latent variables [81,82].

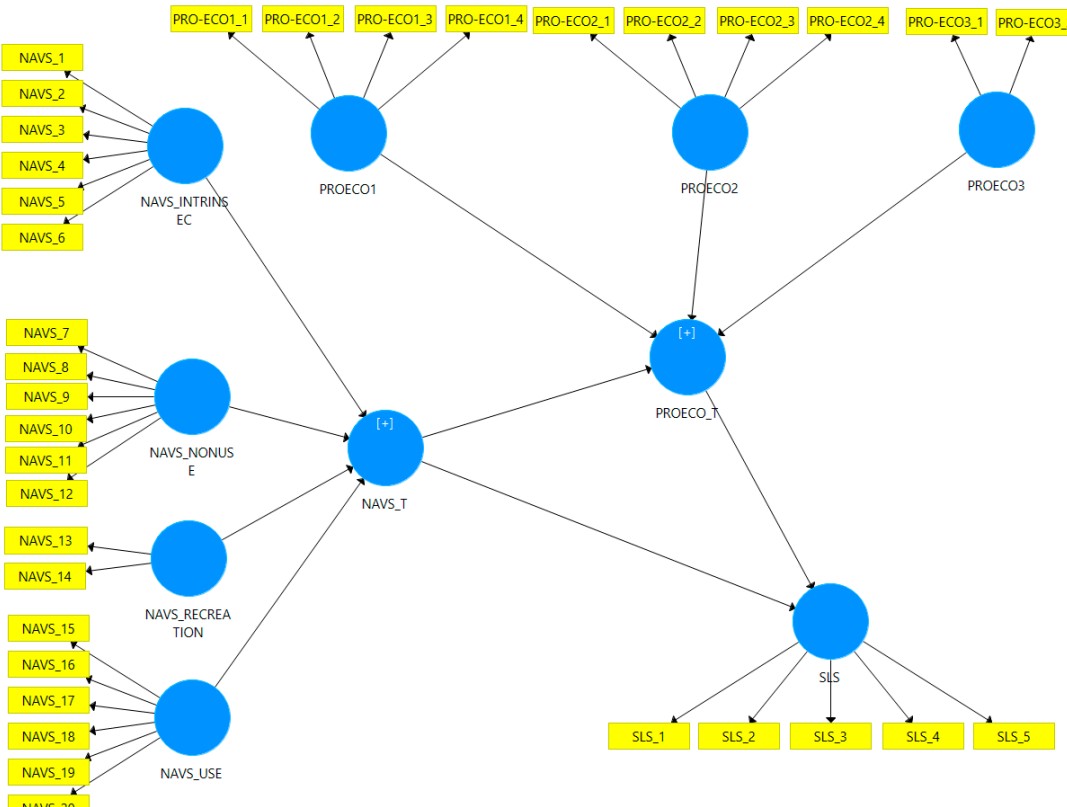

**Figure 2.** Structural model regarding the relationship between the knowledge of environmental values (NAVST), the pro-ecological behaviour (PROECO), and life satisfaction (LSF).

The analyzed model is a hierarchically reflective-formative complex one with reflective and formative variables, with simple and complex latent variables. The use of the PLS method in data analysis with the help of SmartPLS 3.9 software presents a set of advantages for the research:

- it does not assume that data is normally distributed [83];
- the requirements regarding the volume of the batch are more relaxed in the elaboration of a theoretical construct [84];
- it is based on a non-parametric bootstrap procedure [85,86] in order to test the significance of the analysis coefficients of the relations between variables;
- uses an easy visual interface of mental schemas of construct analysis [87] etc.

### 3. Results

1.  The analysis of the conceptual model begins with assessing the significance level of the structural model, by determining the size of the "Cronbach's Alpha" coefficients for the analyzed variables, then of the charge factors, the AVE coefficients, the testing of the variable parameters, etc. [82,87].

    In the case of the model described in Figure 2, after analyzing the statistical indicators, only 18 reflective variables, 5 reflective latent variables, and 2 formative latent variables were retained.

    Following the development, it is ascertained that the research is based on a hierarchical reflective-formative structural model in two stages (higher-order constructs with two-stage) [87–89] using the PLS-SEM method by means of the SmartPLS 3.9. application [63].

1.a  The analysis of the reliability and validity of the reflective model (stage one)

    The analysis of the reliability and validity of the reflective model implies, for a start, the assessment of the level of significance by determining the size of the "Cronbach's Alpha" coefficients and of the reliability coefficients. In the case of the structural model, according to Table 2, the coefficients under discussion have values in the admitted intervals (for the "Cronbach's Alpha" coefficients for the six latent variables, values between 0.726 and 0.784 are obtained, whereas for the Composite Reliability the lowest value is 0.830 and the highest is 0.886) [62,90]. Consequently, it may be asserted that the latent variables preserved in the model are significant (Table 2).

**Table 2.** Measurement model (Stage one).

| Construct | Items | Loading | Cronbach's Alpha | Composite Reliability | Average Variance Extracted (AVE) |
|---|---|---|---|---|---|
| NAVS INTRINSEC | NAVS_1 | 0.762 | 0.727 | 0.830 | 0.549 |
| | NAVS_2 | 0.726 | | | |
| | NAVS_5 | 0.746 | | | |
| | NAVS_6 | 0.731 | | | |
| NAVS USE | NAVS_18 | 0.816 | 0.784 | 0.874 | 0.699 |
| | NAVS_19 | 0.892 | | | |
| | NAVS_20 | 0.797 | | | |
| PRO-ECO1 | PRO-ECO1_1 | 0.764 | 0.755 | 0.845 | 0.577 |
| | PRO-ECO1_2 | 0.698 | | | |
| | PRO-ECO1_3 | 0.796 | | | |
| | PRO-ECO1_4 | 0.777 | | | |
| PRO-ECO2 | PRO-ECO2_1 | 0.902 | 0.744 | 0.886 | 0.796 |
| | PRO-ECO2_2 | 0.882 | | | |
| SLS | SLS_1 | 0.797 | 0.829 | 0.880 | 0.596 |
| | SLS_2 | 0.776 | | | |
| | SLS_3 | 0.815 | | | |
| | SLS_4 | 0.794 | | | |
| | SLS_5 | 0.669 | | | |

Source: Data processed by the authors using SmartPLS 3.9 software, Darmstadt, Germany.

The convergent validity of the model: The first stage involves the analysis of the loading coefficients (in Table 3 the lowest value is >0.669) and of the AVE coefficients (the lowest value is >0.549), and it may be concluded that it is respected [89].

**Table 3.** Fornell-Larcker Criterion—Stage one.

| | NAVS_INTRINSEC | NAVS_USE | PROECO1 | PROECO2 | SLS |
|---|---|---|---|---|---|
| NAVS_INTRINSEC | 0.741 | | | | |
| NAVS_USE | 0.091 | 0.836 | | | |
| PROECO1 | 0.220 | 0.377 | 0.760 | | |
| PROECO2 | 0.143 | 0.481 | 0.409 | 0.892 | |
| SLS | 0.451 | 0.082 | 0.225 | 0.052 | 0.772 |

Source: Data processed by the authors using SmartPLS 3.9 software, Darmstadt, Germany.

Analyzing the 18 reflective variables corresponding to the 5 latent variables from the first stage, the following values of loading, graphically represented in Figure 3, are obtained.

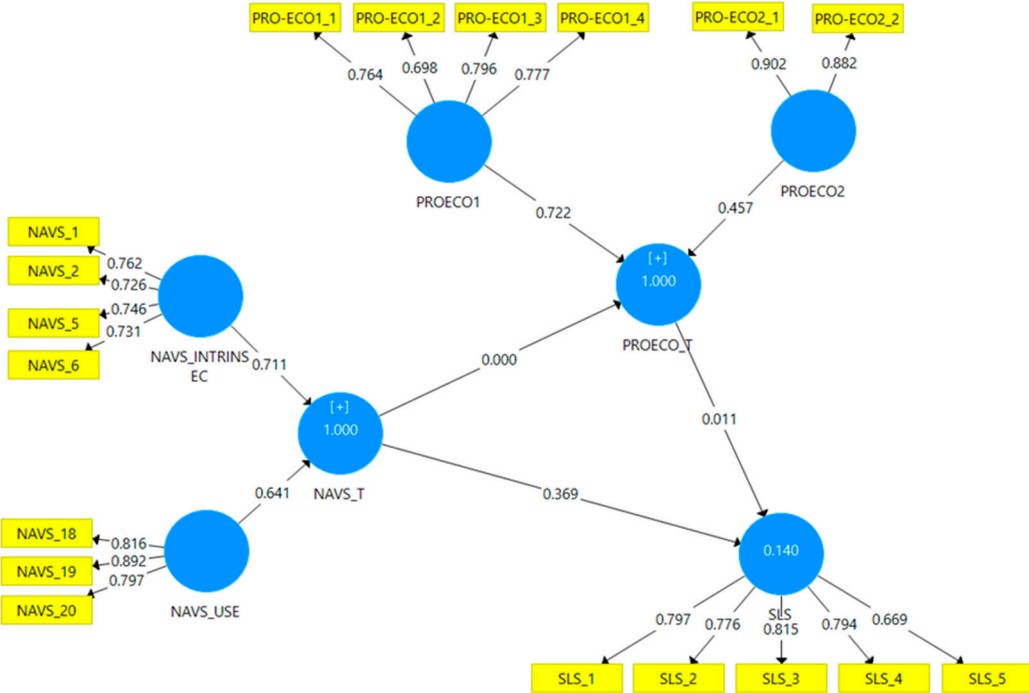

**Figure 3.** Loadings factors.

The discriminating analysis regarding the validity of the model in the first stage is studied by means of the Fornell-Larcker Criterion and the HMTS test.

In Table 3, the values on the main diagonal are significantly higher than the values under the diagonal (the AVE square roots of the latent variables are higher than their correlations with the other variables). Therefore, the theoretical criteria of validity are respected [81].

Following the values in Table 4 regarding the Heterotrait-Monotrait Ratio test, we observe that all values are lower than 0.9, with the validity criteria being respected [61,82].

**Table 4.** Heterotrait-Monotrait Ratio (HTMT)—Stage one.

| | NAVS_INTRINSEC | NAVS_USE | PROECO1 | PROECO2 |
|---|---|---|---|---|
| NAVS_INTRINSEC | | | | |
| NAVS_USE | 0.119 | | | |
| PROECO1 | 0.290 | 0.491 | | |
| PROECO2 | 0.199 | 0.626 | 0.538 | |
| SLS | 0.573 | 0.103 | 0.288 | 0.081 |

Source: Data processed by the authors using SmartPLS 3.9 software, Darmstadt, Germany.

It may be concluded that the discriminated validity of the reflective model in stage 1 (Tables 3 and 4) is respected [87,89].

1.b    The validity of the model for the second-order latent variables (second stage)

The model of analyzing the knowledge of the values in relation to the pro-environmental behaviour and life satisfaction, respectively, among the students in our research, is a hierarchical construct based on four first order components (MAVS_Intrisec, NAVS_Use, Proeco1 and Proeco2) structured in two second-order components (NAVST AND PROECOT).

The analysis of the validity of such a hierarchical construct entails checking the following coefficients: outer weights, outer loadings, and VIF.

In Table 5, we notice that the values for outer weights are significant ($p$ value < 0.05) and the values for outer loadings are higher than 0.5 (ranging between 0.722 and 0.873). Analyzing the size of the VIF interior coefficients, in Table 5 the values are lower than 3, confirming the validity of the construct in the second stage [89].

**Table 5.** Higher-order construct validity (second stage).

| | | Outer Weights | T Statistics (丨O/STDEV丨) | $p$ Values | Outer Loadings | VIF |
|---|---|---|---|---|---|---|
| NAVST | NAVS_INTRINSEC → NAVST_ | 0.665 | 4.696 | 0.000 | 0.728 | 1.008 |
| | NAVS_USE → NAVST_ | 0.688 | 4.732 | 0.000 | 0.749 | 1.008 |
| PROECOT | PROECO1 → PROECOT | 0.654 | 4.565 | 0.000 | 0.873 | 1.200 |
| | PROECO2 → PROECOT | 0.535 | 2.836 | 0.005 | 0.802 | 1.200 |

Source: Data processed by the authors using SmartPLS 3.9 software, Darmstadt, Germany.

2.    Higher-order structural model and the validation of the hypotheses

The relationships analyzed in the construct developed above reflect the following values for testing the parameters described in Table 6.

**Table 6.** Testing the assumed relationships through research.

| | Original Sample (O) | Sample Mean (M) | Standard Deviation (STDEV) | T Statistics (丨O/STDEV丨) | $p$ Values |
|---|---|---|---|---|---|
| NAVST_ → PROECOT | 0.494 | 0.496 | 0.070 | 7.004 | 0.000 |
| NAVST_ → SLS | 0.359 | 0.361 | 0.087 | 4.100 | 0.000 |
| PROECOT → SLS | −0.004 | 0.005 | 0.082 | 0.048 | 0.962 |

Source: Data processed by the authors using SmartPLS 3.9 software, Darmstadt, Germany.

## 4. Discussion

The construct by which the relationships among the three concepts have been analysed is graphically represented in Figure 4. Significant statistical values may be observed only for the connections between the knowledge of the environmental values and the pro-ecological behaviour (NAVST and PROECOT), knowledge of the environmental values, and life satisfaction (NAVST and SLS) [88,89].

**H1** Hypothesis analyzes the relationship between knowledge of the environmental values/concepts (NAVST) and the pro-ecological behaviour (PROECOT). We observe that among the students, knowledge of the environmental concepts has a significant impact on the pro-environmental (pro-eco) behaviour. Consequently, H1 hypothesis is supported ($\beta$ = 0.494, t = 6.629, $p$ values = 0.000).

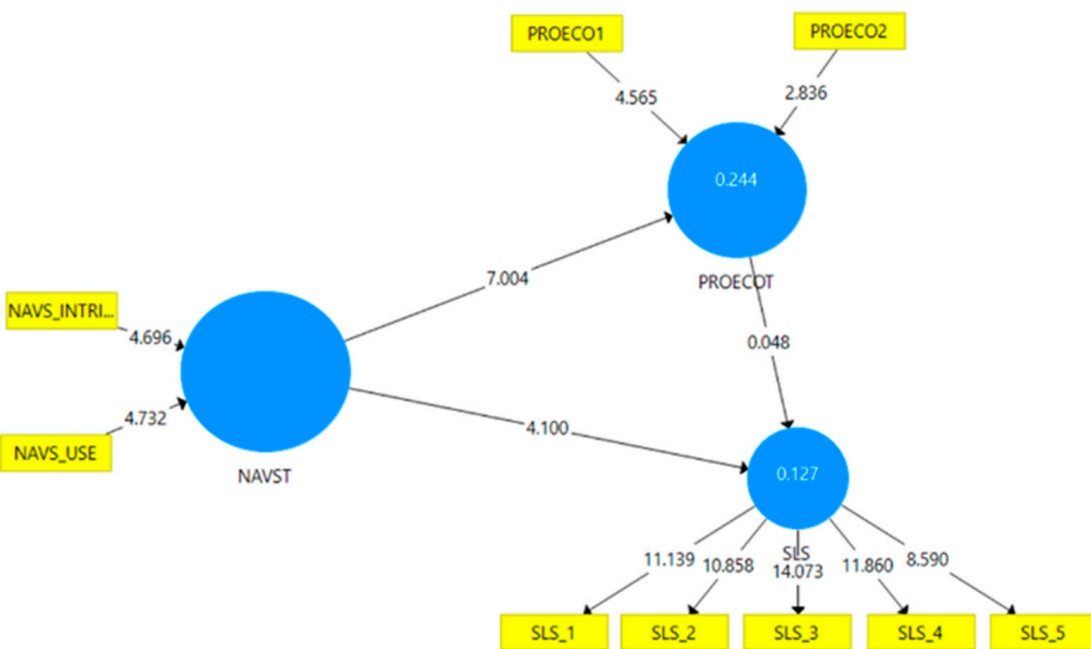

**Figure 4.** Second-stage model between the variables: knowledge of the concepts, favorable behavior, and life satisfaction, among the students.

Within the elaborated construct, we observe that H2 Hypothesis is validated, and the knowledge of the environmental values (NAVST) has a significant impact on life satisfaction (LSF) among the students (β = 0.359, t = 4.021, *p* values = 0.000).

Regarding the H3 hypothesis, the relationship between assuming a pro-ecological behaviour and life satisfaction among the students (SV) is analyzed. It can be observed that the research results do not indicate any influence of PROECOT on LSF (β = −0.004, t = 0.046 and *p* value = 0.963). Therefore, H3 hypothesis is invalidated.

This research stage leads to the analysis of H4 hypothesis. Almost automatically one can speak of the invalidation of this hypothesis, i.e., assuming a pro-ecological behaviour among the students does not bring a significant mediation between knowledge of environmental values and life satisfaction (for the relation NAVST_ → PROECOT → SLS, β = −0.002, t = 0.047, *p* values = 0.962 can be observed). We may conclude that among the students, the influence of knowing the environmental values has a direct, significant impact on life satisfaction. A significant mediation of the relationship at this stage of their development, by assuming a pro-eco behaviour, may not be observed.

Within H5 hypothesis, another research tool is being used: A multi-group analysis using SmartPLS [89,91].

As part of this hypothesis, the students were split into two groups according to the background (urban/rural), in order to check whether there are significant differences within the analyzed relationships.

The utilized analysis uses independent batches of tests t, which allow the comparison of the coefficients between the two groups for all the links described in the construct.

Regarding the relation: knowledge of the environmental concepts → pro-eco behaviour (NAVST_ → PROECOT), in the group of students from the rural area (β = 0.538, t = 9.211, *p* values = 0.000), as well as in those from the urban area (β = 0.421, t = 3.363, *p* values = 0.001), the conservation of the significant impact described in H1 hypothesis may be observed. Moreover, slightly higher values of the opinions among the students from the rural area may be noticed, but the differences are not statistically significant (*p*-value >> 0.05, Table 7). The same situation is observed for the relation NAVST_ → SLS (knowledge of the environmental values → life satisfaction).

**Table 7.** The results of the multi-group analysis (MGA, GR_Rural(2.0)-GR_Urban(1.0)).

| Links | Path Coefficients-diff | *t*-Value New | PLS-MGA New | Parametric Test | Welch-Satterthwait Test |
| --- | --- | --- | --- | --- | --- |
| | | | *p*-Value | *p*-Value | *p*-Value |
| NAVST_ → PROECOT | 0.138 | 0.87 | 0.302 | 0.383 | 0.345 |
| NAVST_ → SLS | 0.075 | 0.40 | 0.652 | 0.688 | 0.687 |
| PROECOT → SLS | −0.056 | 0.31 | 0.738 | 0.75 | 0.749 |

Source: Data processed by the authors using SmartPLS 3.9 software, Darmstadt, Germany.

The obtained results presented in Table 7 highlight the lack of a statistically significant difference between the students from the urban area and the group from the rural area (*p*-value indicates values between minimum 0.302 and maximum 0.749) [87,92,93]. Thus, we may conclude that H5 hypothesis is validated. For the students from the rural area, slightly higher values of their opinions compared to those from the urban area may be observed; however, from a statistical point of view, there are no significant differences of the opinions recorded among the two groups.

From our point of view, on the one hand, the researched population is one with a high level of education that assimilates and implements more easily sustainable concepts favorable to the protection of the environment, while on the other hand, a high educational process leads to attitudes that are significantly similar and favorable to the protection of the environment.

From the organization of the paper point of view, the models proposed for analysis were validated in two stages. Both the validation of the constructs and the analysis of the reliability of the models were carried out. Within our research, the five assumed hypotheses were analyzed by means of a complex hierarchical reflective-formative model, with 35 reflective variables, 8 reflective latent variables, and 2 formative latent variables, with simple and complex latent variables focused on the three main constructs: knowledge of the environmental values (NAVS), pro-ecological behaviour (PROECO), and life satisfaction (LSL) among the students.

The initial conceptual model was validated only for a part of the reflective items and the latent variables, being preserved in the analysis for the knowledge of the concepts related to the dimensions of intrinsic values and non-material values, and for those related to behaviour—the components regarding attitudes and the actual behaviour.

The performed analyses using the PLS-SEM method emphasized the fact that between the knowledge of the environmental concepts and the pro-environmental behaviour, there is a statistically significant relationship among the students (H1 hypothesis), as well as between the knowledge of the environmental concepts and life satisfaction (H2 hypothesis). A significant connection between behaviour and life satisfaction could not be demonstrated (H3 hypothesis). Furthermore, it has been found that among the students from the rural area, slightly higher values of the opinions were identified compared to those from the urban area, even if they are statistically insignificant (validating the H5 hypothesis). Because the environmental values are treated relatively little in the relevant literature, especially domestic ones, and that ecological behavior is predominantly referred to, we highlighted the importance of knowing environmental values for satisfaction with life. Environmental values, ecological behavior, and life satisfaction, although they are concepts from various fields, are strongly interrelated. Knowledge of environmental values is found to be essential to life satisfaction. For a better understanding of the requirements of sustainable development, it is first necessary to deeply know the environmental values that allow, out of conviction, the orientation towards ecological behavior. This is the way to appreciate, at their fair value, the resources of the environment.

## 5. Conclusions

Based on the results obtained from the survey, by analyzing and processing the data and by considering the formulated working hypotheses, the following specific conclusions

are highlighted: the three analyzed concepts (environmental values, pro-eco behavior, and life satisfaction) are interdependent. The concepts have practical significance for the academic environment, with reference to students enrolled in specializations aimed at environmental and natural resources issues. The "knowledge of environmental values" component is a solid and significant one for stimulating a sustainable behavior manifested by caring for the environment and protecting its values. So, it is essential for students to know and appreciate environmental values, both conceptually and taxonomically. The "knowledge" component, with reference to environmental values, has a strong and significant impact on life satisfaction among students. Thus, the more thoroughly students know theories, concepts, and notions about the environment and its values, the more satisfied they are with their personal lives. One of the surprising conclusions, based on the results achieved, is that a "pro-eco" behavior has no direct influence on life satisfaction, so students can be satisfied with their lives only if they have knowledge about environmental values, so not necessarily if they show a pro-eco behavior. Moreover, assuming a pro-eco behavior does not improve the link between knowledge of environmental values and life satisfaction, the latter are dependent on each other, without considering the impact of pro-eco behavior.

It can be concluded that among students, the influence of knowledge of environmental values has a significant impact on life satisfaction, directly. From the gender differences point of view (female vs. male), there are no significant differences for the analysis. On the contrary, with regard to the urban-rural differences, slightly higher values of the opinions are found among students from rural areas. The differences are not statistically significant, but may have, as a causal explanation, the training environment through more direct contact with environmental factors and values. So, in terms of value, there is an intrinsic component of the appreciation of the environment with the resources available to it, by rural students, regardless of gender. We can also include the "knowledge of environmental values—satisfaction with life" relationship here; the same difference is found, without statistical significance in particular: Students from rural areas are more satisfied with life, through knowledge of environmental values, compared to students from urban areas. Otherwise, there are no significant differences between the two-reference means.

The connection between the three concepts, reinforced by these conclusions, as the novelty of the research, indicates the following three different meanings: cognition influences behavior and life satisfaction, behavior influences cognition, and life satisfaction influences cognition and behavior. Moreover, young people are increasingly concerned about environmental and resource problems, not from an extrinsic perspective, but from that of their own person. The issue of the environment, with its values, in association with sustainable development policies, is transferred from external and material aspects toward personal concerns. An ecological behavior manifested in the context of not understanding the environmental values intrinsically does not determine long-term results and is superficial. In addition, in this regard it is found that the knowledge of environmental values is essential and has a significant influence on satisfaction with life, and the ecological behavior has no direct influence.

By proposing recommendations and measures applicable in various practical environments (academic, entrepreneurial, legislative, and psycho-emotional), the following elements will be taken into account: knowledge of environmental values and satisfaction with life. Behavior remains the element that can be most easily adapted, if the other two elements are solid. For the most significant recommendations we developed, increased attention is paid to the learning and entrepreneurial environment. Also, it is recommended to focus on providing information about environmental values in order to know them, because the knowledge of environmental values and the application of knowledge within an ecological behavior are vectors of life satisfaction. A proposal is the focusing on formal education for the purpose of more in-depth knowledge of the meaning of environmental values and consideration of sustainable development measures in multiple fields and at the individual level, including on an emotional level. The results of this research can be better exploited, compared to the legislative or psycho-emotional environment. Thus, it is

recommended to emphasize the accumulation of knowledge about environmental values in an organized learning framework, through a reconsideration of an appropriate curriculum. Both students and pupils need to gain a better understanding of why it is important to be informed about environmental values. Also, young people must learn and understand that by knowing and recognizing environmental values, their life satisfaction will be higher. For now, in native learning environments, this aspect is not covered well enough. In this sense, behavior can prove to be a usable tool by educational institutions in order to stimulate knowledge. For the entrepreneurial environment, the results of this research can prove their usefulness by considering partnerships with educational institutions. By creating an appropriate framework for environmentally responsible activities, companies can stimulate pro-eco behavior and increase the quality of life in multiple and varied ways. Likewise, the entrepreneurial environment can realize innovative projects based on the connection and impact between environmental values and satisfaction with life.

The work was supported by the consistency of relevant, especially international, literature. Interdisciplinary relevance and importance were also argued. The proposed objectives were achieved, as follows: the influence of knowledge of environmental values on the well-being of the individual was analyzed in detail, the influences of social differences on ecological behavior were identified, and ecological behavior was analyzed from the impact on the well-being point of view.

The main added value of the paper consists in creating a hierarchical conceptual model for the analysis of sustainability based on constructs such as environmental values and pro-ecological behavior, as well as satisfaction with life. The model was only partially validated, being highlighted: the existence of significant equations between knowledge of environmental values and pro-ecological behavior among students, and between the knowledge of environmental values and satisfaction with life. The relationship between pro-ecological behavior and life satisfaction among students from Iasi was invalidated. Also, this reflective-formative model of structural equations on two hierarchical levels, partially validated in the research, does not vary depending on the subject's rural and urban area. Concretely, we can support interdisciplinarity and start structuring and defining a new conceptual model based on the results obtained; education remains vital for the assumption and recognition of environmental values. Very emphasised is the interconnection of apparently unrelated fields of study and the modeling of the relationship between the three concepts under the conditions of an interdisciplinary approach. In addition, sustainable development is shown to span several different fields of study. Identifying a conceptual model that integrates the three concepts in an interdisciplinary manner, and highlighting the fact that it is not the ecological behavior that prevails in satisfaction with life, but knowledge of environmental values, is the strong point of the paper. This model can be extended to a much larger population with an emphasis on other specific dimensions, and for which new elements can be added within the established questionnaires, regarding behavior, knowledge of values, and satisfaction with life.

However, this research has a set of limits: The analyzed sample was limited to students from the Municipality of Iași, so an extended analysis is required if the generalization of the results at the national level is desired. The research only focused on college students, but expansion of the study to the youth population, regardless of whether or not they attend college, is being considered. The research considered only students from specialized faculties who already have information and affinities regarding environmental values, so, the results cannot be generalized to all young people. In addition, the sample consists only of young people. The subjectivity of the choice of working concepts and variables can prove to be another limitation, along with the difficulty of selecting the most suitable scales for the purpose of this research. In addition, although the literature is rich in the individual approach to the three analyzed concepts (environment, ecological behavior, and life satisfaction), more numerous and relevant sources could have been identified for bringing these concepts together. As future solutions for these limits: Future research should be conducted on a sample of students from all over the country and from other

specializations; a comparative analysis could be carried out on university centers and specializations.

Consequently, we aim to expand research on a regional or even national scale in the future and redefine the research universe. Also, a concept with high influence potential will be added: the personality of the individual. Using a qualitative method of analysis, such as "top-of-mind" or the "free association", could complement the results of future research with high consistency. Last, but not least, it could be considered a dynamic analysis, based on a longitudinal recording data set.

This result leads to the further development of research on the sustainability of consumption on a much larger population in terms of area and diverse categories, an approach allowing to analyze much more complex dimensions within the three internal constructs, and also including moderating aspects such as age, level of training, and so on. In conclusion, environmental values, pro-eco behaviors, and life satisfaction are shown to be in a significant interdependent relationship. Students, as the targeted research universe, can exhibit pro-environmental behaviors without increasing their level of satisfaction with life; instead, by accumulating knowledge and information about environmental values, they can be more satisfied, and the more satisfied they are in life, the more open they are to knowledge and learning about the environment, natural resources, and sustainability. All this can be achieved, if a formal, but also informal learning framework is provided.

**Author Contributions:** Conceptualization, A.F.J., M.M. and C.-I.A.; methodology, C.-I.A., A.-D.R. and G.I.; software, C.-I.A. and A.-D.R.; validation, G.I. and C.L.C.; formal analysis, A.F.J., M.M., C.-I.A. and A.-D.R.; investigation, M.M. and C.-I.A.; resources, A.F.J., M.M. and C.-I.A.; data curation, A.-D.R., G.I. and C.L.C.; writing—original draft preparation, A.F.J., M.M., C.-I.A., A.-D.R., G.I. and C.L.C.; writing—review and editing, M.M. and A.-D.R.; visualization A.F.J., A.-D.R., G.I. and C.L.C.; supervision, G.I. and C.L.C.; project administration, A.F.J., M.M. and C.L.C. All authors have read and agreed to the published version of the manuscript.

**Funding:** This research received no external funding.

**Institutional Review Board Statement:** The research conducted was approved by Iași University of Life Sciences Research Ethics Committee (IULS Ref No: 14673/18.08.2022).

**Informed Consent Statement:** Informed consent was obtained from all participants involved in the study.

**Data Availability Statement:** In accordance with the consent provided by participants on the use of confidential data, data are not available to be shared.

**Conflicts of Interest:** The authors declare no conflict of interest.

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
