# Peer review of "The Relationship between Environmental Factors, Satisfaction with Life, and Ecological Education: An Impact Analysis from a Sustainability Pillars Perspective"

_sustainability, doi:10.3390/su141710679_

Round 1
Reviewer 1 Report
Dear authors,
Your study is interesting and well performed, but there are some shortcomings.
Please find my detailed comments below:
1) In the abstract, you need to include 1) the overall purpose of the study and the research problems you investigated; 2) the basic design of the study; 3) major findings or trends found as a result of your analysis; and, 4) a brief summary of your interpretations and conclusions. At the moment, some of these aspects are missing. As an example, lines 21 – 22, “The main results: knowledge of environmental values has a significant impact on life satisfaction” are very general.
2) In the introduction, make clearer what knowledge gaps you identified and how your research addresses them. You present main pursued objectives (lines 294 – 310) but a “so what?” question still remain. A deeper analysis is needed. Can the paper contribute to the existing international literature?
3) I propose you to discuss more the limitations of your research and how you overcome them. Especially the fact that the analyzed sample was only from the university environment in Iași.
I hope these comments are helpful and will improve the manuscript.
Kind regards,
Reviewer 2 Report
It is suggested to highlight the main contribution regarding current knowledge in the abstract section.
State of the art and key literature: It is needed to be much more specific and present a literature review that highlights the gap that the paper aims to address. It is a key question to define what is considered life quality since it can be approached differently. It should be based on specialized literature to avoid any misunderstanding of this concept. A more clear state of the art would be recommended to be done in the introduction section before introducing the paper’s main aim.
Also, the main concepts and ideas must be introduced into a coherent and updated structure in a theory section.
Some paragraphs are repeated throughout the text.
Methods: This section seems too large. A better summary is recommended. It should be interesting to specify why the pretested questionnaire ensures the obtained outcomes and the survey’s informants. It should be better defined in this section.
On the other hand, a survey using the Likert scale explicitly measures attitudes, but it may rise an issue: What about the implicit measures to avoid existing stereotypes or prejudices? There is a huge risk to fall into reductionism if there is a lack of investigation of the social background and cultural influence on environmental values since culture teaches how to feel. Therefore, this lack does not allow knowing if this method would produce the expected results.
Minors mistakes: The point should be after the mentioned reference.
Reviewer 3 Report
Please carefully justify and explain on all comments. The title of the paper should not be capitalised.
1. In the introduction, you need to connect the state of the art to your paper goals. Please follow the literature review by a clear and concise state of the art analysis. This should clearly show the knowledge gaps identified and link them to your paper goals. Please reason both the novelty and the relevance of your paper goals. Clearly discuss what the previous studies that you are referring to.
What are the Research Gaps/Contributions? Please note that the paper may not be considered further without a clear research gap and novelty of the study.
2. Please underscore the scientific value-added to your paper in your abstract. Your abstract should clearly state the essence of the problem you are addressing, what you did and what you found and recommend.
That would help a prospective reader of the abstract to decide if they wish to read the entire article.
3. In your discussion section, please link your empirical results with a broader and deeper literature review.
4. Please make sure your conclusions' section underscores the scientific value-added of your paper, and/or the applicability of your findings/results. Highlight the novelty of your study.
In addition to summarising the actions taken and results, please strengthen the explanation of their significance. It is recommended to use quantitative reasoning comparing with appropriate benchmarks, especially those stemming from previous work.
5. Please consult the journal's reference style for the exact appearance of these elements, and use of punctuation and capitalisation.
Bibliography style is not always consistent, please check the reference section carefully and correct the inconsistency. Please update your introduction with the following references:
https://doi.org/10.1016/j.jclepro.2020.125575
https://doi.org/10.1016/j.bcab.2019.101234
https://doi.org/10.3390/en14092429https://doi.org/10.3390/pr9091629
6. The author needs to draw the graphical abstract by considering that the information in the graphical abstract is informative and reflects the content and results of the research that has been done.
7. The author needs to rearrange the abstract and conclusion according to the key findings concluded remarks.
Round 2
Reviewer 1 Report
The authors have improved the manuscript. The paper can be accepted in present form.
Reviewer 2 Report
All the suggestions have been incorporated into the paper.